# Chemical diversity in a metal–organic framework revealed by fluorescence lifetime imaging

Waldemar Schrimpf[1], Juncong Jiang[2], Zhe Ji[2], Patrick Hirschle[1], Don C. Lamb [1], Omar M. Yaghi [2,3] & Stefan Wuttke [1,4]

The presence and variation of chemical functionality and defects in crystalline materials, such as metal–organic frameworks (MOFs), have tremendous impact on their properties. Finding a means of identifying and characterizing this chemical diversity is an important ongoing challenge. This task is complicated by the characteristic problem of bulk measurements only giving a statistical average over an entire sample, leaving uncharacterized any diversity that might exist between crystallites or even within individual crystals. Here, we show that by using fluorescence imaging and lifetime analysis, both the spatial arrangement of functionalities and the level of defects within a multivariable MOF crystal can be determined for the bulk as well as for the individual constituent crystals. We apply these methods to UiO-67 to study the incorporation of functional groups and their consequences on the structural features. We believe that the potential of the techniques presented here in uncovering chemical diversity in what is generally assumed to be homogeneous systems can provide a new level of understanding of materials properties.

[1] Department of Chemistry and Center for NanoScience (CeNS), University of Munich (LMU), Butenandtstrasse 11, Munich 81377, Germany. [2] Department of Chemistry, University of California-Berkeley, Berkeley National Laboratory, Berkeley, CA 94720, USA. [3] King Abdulaziz City for Science and Technology (KACST), Riyadh 11442, Saudi Arabia. [4] School of Chemistry, Joseph Banks Laboratories, University of Lincoln, Lincoln LN6 7TS, UK. Correspondence and requests for materials should be addressed to O.M.Y. (email: yaghi@berkeley.edu) or to S.W. (email: stefan.wuttke@cup.lmu.de)

The central goal of materials science has always been the synthesis and characterization of materials with novel properties. Key aspects that have an impact on the properties of the material are the presence and distribution of functional groups and defects within the substance[1–6]. This chemical diversity is an unpredictable outcome of chemical synthesis conditions and arises prominently in the chemistry of metal–organic frameworks (MOFs), where the use of multiple functionalized organic linkers results in a multivariable system in which the spatial arrangement of both functionalities and defects are unknown[7–9].

Deciphering the chemical diversity in MOF crystals is an experimental challenge, because most characterization techniques rely on measuring the averaged properties of a bulk sample, such as elemental analysis, powder X-ray diffraction (PXRD), and gas adsorption isotherms. Recently, solid-state nuclear magnetic resonance (NMR) coupled with computational modeling has been shown to be extremely powerful in elucidating the distribution of different linkers and functional groups[10–13]. However, even these measurements are based on statistical averaging of resonances emanating from many different crystals within the sample. Single-crystal X-ray diffraction, on the other hand, examines the chemical nature of defects in only one single crystal at a time[14]. Therefore, neither one of these methods provides information about the diversity of the sample as a whole. Electron microscopy, although extensively used to map defects in inorganic solids, is problematic for MOF imaging as the electron doses required to image the material with good resolution quickly damage the sample. Still, low-dose transmission electron microscopy has been very recently used to study surfaces and interfaces of the MOF ZIF-8[15]. In contrast, fluorescence imaging has the advantages of spatial resolution, high throughput, and sensitivity. It was previously employed to visualize plane defects and surface functionalization in MOFs[16], and to compare the linker distribution between different methods of synthesis[13]. Extending the measured fluorescence parameters beyond the intensity, e.g., to the spectrum[13] or the lifetime[17, 18], greatly increases the information gathered about the material. Most fluorophores are highly sensitive to their immediate surroundings, so that changes in the nanoscopic environment affect their photo-physical properties, including the fluorescence lifetime. By careful analysis of the lifetime, it is therefore possible to detect and interpret spatial heterogeneities or differences between samples on length scales far below the resolution limit of a light microscope, thereby complementing the insights gained using other imaging techniques, such as electron microscopy or Raman imaging[19].

In this study, we apply fluorescence imaging combined with fluorescence lifetime analysis to examine the diversity and distribution of defects and functional groups in a MOF. Fluorescent dye modified linkers were incorporated into the UiO-67 framework, serving as both model functional group and reporter. Förster resonance energy transfer (FRET) analysis indicates a random distribution of the incorporated dyes, whereas fluorescence lifetime imaging (FLIM) revealed a correlation between fluorescence quenching and nanoscopic defects, aspects not detectable with standard bulk characterization techniques. The measurements uncovered chemical diversity in a multivariable MOF originating from different synthesis conditions, within a sample, and even within a single crystal, highlighting the potential of fluorescence based methods in decoding the state of complex porous materials.

## Results

**Bulk characterization**. UiO-67 was chosen as a MOF prototype for this study, because it exhibits exceptional chemical and thermal stability, and has pores large enough to incorporate the dyes used for functionalization[20]. In this MOF, $Zr_6(\mu_3\text{-O})_4(\mu_3\text{-OH})_4$ clusters are connected to 12 linear ditopic organic linkers (biphenyl-4,4′-dicarboxylic acid, BPDC), forming a network of face-centered cubic topology. A portion of the original linkers can be substituted with dye-modified versions, creating isoreticular structures without altering the topology. For this purpose, the organic linker 2-amino-BPDC was coupled to either fluorescein isothiocyanate (FITC) or rhodamine B isothiocyanate (RITC). These dye-functionalized linkers were then incorporated into the framework either de novo (i.e., added to the synthesis solution) or by postsynthetic linker exchange[21, 22]. Using the lowest MMFF94 energy conformations, the maximal projection radii of the dyes attached to the linker were estimated to be 18.1 Å for FITC and 20.5 Å for RITC. This makes the dyes small enough to fit into the 23 Å wide octahedral pores of UiO-67, but not into the 11.5 Å large tetrahedral pores[23]. If the linkers are incorporated into the framework itself, the sizes of the fluorophores protruding into the pores are 14.3 Å and 17.9 Å, respectively, allowing for a better fit.

The total amount of dye-functionalized linker incorporated into the scaffold was determined by comparing the measured fluorescence signal of the digested MOFs in an aqueous solution in relation to the total amount of linker in the sample. The resulting de novo incorporation efficiencies (i.e., the ratio of the incorporated and the input fractions) were 17–27% for FITC and 2–7% for RITC (Supplementary Table 1). It is likely to be that this difference is caused by the larger side groups of RITC and its positive charge. The crystallinity, porosity, and morphology of the dye-modified MOFs were investigated with PXRD (Supplementary Fig. 1), nitrogen adsorption/desorption isotherms (Supplementary Table 2), and scanning electron microscopy (SEM, Supplementary Fig. 2), respectively. These bulk measurements detected no changes caused by the functionalization, independent of the incorporation method, or the type and the amount of incorporated dye.

**Distribution of functional groups**. In order to determine the distribution of functional groups in the UiO-67 framework, two-color experiments were performed by incorporating both dyes into the MOF using de novo functionalization. The microscopic distribution of fluorophores is given from the fluorescence intensities within the MOFs. The nanoscale distribution of the dyes can be investigated using FRET as the fluorophores constitute a FRET pair with FITC serving as the donor and RITC as the acceptor dye. During FRET, energy is transferred nonradiatively from the excited donor fluorophore to the acceptor. This process results in quenching of the donor fluorescence, indicated by a decrease in the fluorescence lifetime, and a corresponding increase of the signal in the acceptor channel. The FRET efficiency is highly distance dependent ($R^6$ dependence) and can be exploited to measure the separation between fluorophores on the nanometre scale. Previous studies have already used this property in combination with FLIM to investigate the spatial distribution of fluorophores in MOFs and other porous materials[18, 24, 25]. The Förster radius, representing the dye separation resulting in 50% transfer efficiency, is in the range of 40–80 Å for common fluorescence dyes. It depends on the particular dye pair and the measurement conditions, most notably the spectral overlap integral between the donor emission and the acceptor absorbance, the donor fluorescence quantum yield, and the relative orientation between the dyes. Based on these properties, the Förster radius of the dye pair of FITC and RITC in water can be calculated as 55 Å. However, inside the MOF pores, these parameters can be affected in an unpredictable way, changing the Förster radius. A key factor hereby is the relative

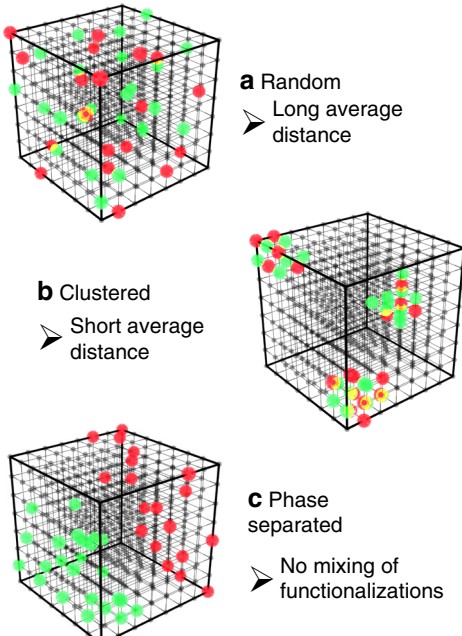

**Fig. 1** Schematic representation of the possible spatial distributions of two different functionalities within the MOF structure. **a** A random distribution with a long average separation between functional groups, green and red. **b** A clustered distribution with short average distances between functional groups. A sub-form of clustering is a distribution at or close to the surface. **c** A phase-separated distribution with very long distances between different functional groups. In the different phases, the individual groups themselves can be both clustered or random

orientation between the FRET pair. Inside of the pores, it is very likely to be that the rotation of the fluorophores is restricted, resulting in a preferred orientation for each individual pair of dyes. On the other hand, the structure of UiO-67 shows high symmetry, so that the individual molecules will assume several orientations. Averaging over multiple fluorophores should give a similar result as for freely rotating dyes. Although the uncertainty regarding the Förster radius inside of the MOF makes a quantitative analysis of the inter-dye distances difficult, the presence or absence of FRET can still be used to judge the general distribution of the fluorophores.

For these investigations, the fraction of FITC was kept constant at 0.1% of the linkers during synthesis, whereas the amount of RITC was varied between 0.01% and 1%. Based on the incorporation efficiency of RITC, this concentration range should result in long average distances between the dyes and therefore produce no significant FRET signal, assuming a purely random and homogeneous nanoscale distribution of the dyes in the scaffold (Fig. 1a). However, clustering, either at the surface or inside of the crystal, would create regions with higher fluorophore densities (Fig. 1b), thus lowering the nearest-neighbor distances, and result in measurable FRET. From this, the presence and degree of clustering can be determined.

The particle shapes observed with fluorescence imaging (Fig. 2) correspond well to the morphology revealed by SEM (Supplementary Fig. 2) for all measured samples. Furthermore, a significant, homogeneous fluorescence signal was observed for both dyes in all MOF crystallites (Supplementary Fig. 3), indicating that FITC and RITC are well mixed within the resolution limit of the microscope (~ 200 nm). Besides the intensity, the fluorescence lifetime of the MOFs was analysed using the phasor approach. Unlike standard fit-based lifetime analysis approaches, phasor FLIM is calculated by simple

mathematical equations and is therefore not biased by the selected fit models. Instead, it uses the Fourier space to visualize the measured fluorescence lifetime in a graphical way[26, 27]. This makes it very useful for a qualitative analysis of complex data with many unknown processes and contributions, as is the case for the presented data[17]. Detailed descriptions of the phasor calculations and rules are given in the supporting information and previous publications[17, 26]. In short, the first cosine ($g$) and sine ($s$) Fourier coefficients of the fluorescence decay (Fig. 2a) are calculated for each pixel of the corresponding FLIM image and plotted as a two-dimensional histogram (see Fig. 2b). Hereby, all purely mono-exponential decays lie on a semi-circle of radius 0.5 around the point (0.5, 0), with lifetimes decreasing in the clockwise direction. Multi-exponential decays can be treated as a vector addition of the base components and will fall inside of that circle. However, due to shot noise and the corresponding inaccuracy, the phasor of individual pixels can fall outside of the circle In this study, we plot the right-hand side of the phasor plots (i.e., the semi-circle with $0.5 < g < 1$). This means that a decrease in lifetime corresponds to a shift from the top left of the plot to the bottom right (Fig. 2b).

The phasor analysis shows a clear decrease of the FITC lifetime with increasing RITC concentration, from $2.07 \pm 0.18$ ns at 0.01% input RITC fraction, down to $0.83 \pm 0.26$ ns at 1% RITC input fraction. FRET has been previously reported as one possible quenching source for dyes incorporated into the UiO-67 scaffold[28, 29]. However, a detailed analysis of the data suggests a different explanation. The RITC signal after 475 nm laser excitation (used for FITC excitation) shows the identical decay as the RITC signal after 565 nm excitation (used for direct RITC excitation), even at the highest RITC concentration (Fig. 2h–j). In the case of FRET, the donor is excited first and then transfers the energy to the acceptor, leading to a delay in the fluorescence lifetime of RITC. As this is not observed, the RITC signal after 475 nm excitation is mainly caused by direct RITC fluorescence. If FRET occurs, it is, at most, a minor contribution to the fluorescence quenching of FITC and some other mechanism must be present. A similar decrease in lifetime with increasing dye concentration was also observed for samples containing only a single fluorophore type (Supplementary Fig. 5) that is also associated with a small spectral shift (Supplementary Fig. 6). Although FRET between identical fluorophores is also possible, it does not directly result in a shorter lifetime and requires significantly higher dye densities to cause quenching. This further supports the argument against FRET between fluorophores as the main source of quenching.

Based on the total amount of incorporated dye in the sample with 0.1% FITC and 1% RITC, the average distance to the nearest RITC dye, assuming a purely random distribution (Fig. 1a), was calculated to be 63.6 Å, a value close to the FRET range. Clustering would result in a significant decrease of the nearest-neighbor distance (Fig. 1b), thus causing energy transfer to occur, even when accounting for possibly shorter Förster radii inside the MOF. Thus, the absence of FRET also means the absence of clustering. At the same time, the decrease in FITC lifetime due to the incorporation of RITC indicates some interaction between the two fluorophores. As no large-scale changes were observed in the bulk characterization methods or in the fluorescence intensity, this affect must be localized within a few tens of nanometers. Thus, we can preclude phase separation (Fig. 1c), suggesting that the dyes are distributed purely randomly in the framework.

**Defects in the scaffold quench fluorescence.** As FRET is the dye–dye interaction with the longest range, the absence of energy transfer suggests that a direct interaction between the fluorophores is not the source of the fluorescence quenching. At the

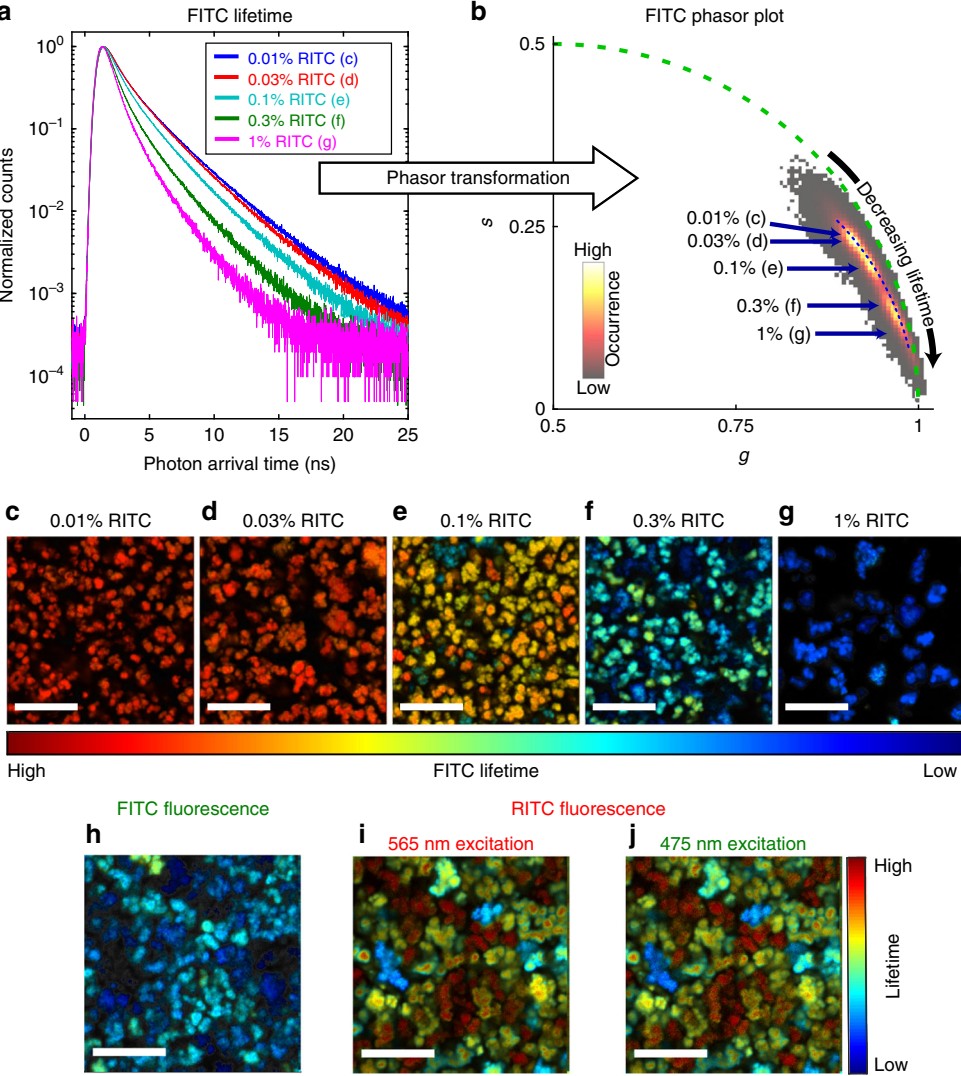

**Fig. 2** A phasor analysis of UiO-67 samples with both FITC- and RITC-labeled linkers. The fraction of labeled linkers during sample preparation was held constant at 0.1% for FITC, whereas for RITC it was varied from 0.01% to 1%. **a** Photon arrival time histogram of the FITC fluorescence for the full images shown in **c–g**. **b** The phasor histogram calculated from the pixel wise photon arrival time histograms of FITC of the images shown in **c–g**. The blue arrows point at the average phasor positions of the different samples. The green dotted line is the universal circle, indicating the possible positions of single exponential decays. **c–g** Lifetime images of FITC fluorescence of UiO-67 samples with 0.1% input FITC linker fraction and 0.01% (**c**), 0.03% (**d**), 0.1% (**e**), 0.3% (**f**), or 1% (**g**) input RITC linker fraction. The color code represents the pixel phasor positions along the blue dotted line in **b**. **h–j** FLIM images for different excitation and detection schemes of the UiO-67 sample with 0.1% FITC and 1% RITC input fraction. The corresponding phasor plot is shown in Supplementary Fig. 4. **h** FLIM image of the green detection channel (500–540 nm) after 475 nm excitation, corresponding to the direct FITC fluorescence. **i** FLIM image of the red detection channel (570–620 nm) after 565 nm excitation corresponding to the direct RITC fluorescence. **j** FLIM image of the red detection channel after 475 nm laser excitation representing the possible FRET signal. The scale bar in all images is 10 μm

same time, all conditions, with the exception of the fraction of dye-modified linkers, were kept constant during synthesis, meaning that the observed changes must be caused by the fluorophores themselves. The only possible explanation consistent with both observations—quenching of FITC by RITC incorporation, but no FRET—is that dye incorporation results in changes in the scaffold itself. The most likely explanation is that incorporation of dye-functionalized linkers during the formation of the MOF interferes with the crystal growth, resulting in defects. It is easy to imagine that the big and bulky dyes can sterically hinder proper scaffold formation. In addition to this steric effect, both fluorophores also have an additional carboxyl group that can interact with the zirconium clusters, further obstructing correct growth of the framework, an interference resulting in vacancies or mismatches in the crystal structure[30–33]. Considering the existing

literature on zirconium-based MOFs[14, 34–37], almost all types of crystal defects—missing linkers, missing clusters, or lattice mismatches—can result in coordinatively unsaturated metal sites (CUSs), and thus lead to fluorescence quenching, as has been shown earlier[17, 38]. Higher dye concentrations create more defects and thus lead to more quenching. The fact that the morphology, the crystallinity, and the pore volume are unaffected indicates that these defects are very localized, extending only on the nanometer scale.

**Measuring the chemical diversity between particles.** Using both the spectral and the lifetime information, it is possible to determine the functionalization and defect level not only for the whole sample, but also for individual particles. Both the amount of dyes

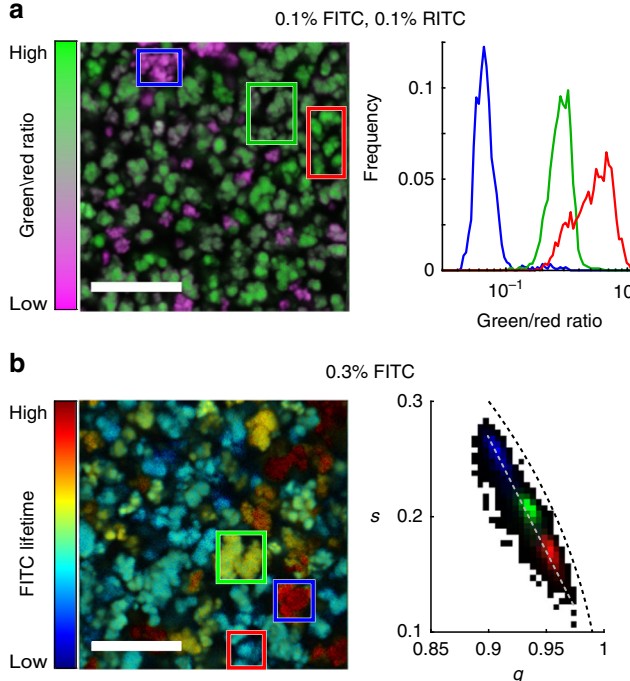

**Fig. 3** Fluorescence data for individual UiO-67 particles. **a** An image (left) and histogram (right) showing the ratio of FITC to RITC fluorescence of a UiO-67 sample with 0.1% FITC and 0.1% RITC linker input fraction. The three curves represent the regions selected with the coloured rectangles in the image. **b** A phasor FLIM image (left) and histogram (right) of the UiO-67 sample with 0.3% FITC linker input fraction. The 2D colored histograms correspond to the regions highlighted with the colored rectangles in the image. The color coding of the images is based on the pixel phasor position along the gray line in the phasor histogram. The scale bar in the images is 10 μm

(Fig. 3a), as well as the level of defects (Fig. 3b), vary significantly between individual MOF aggregates, whereas the diversity within a single particle is much smaller. For example, the SD of the pixel lifetimes for the full image shown in Fig. 3b is 0.21 ns. However, for individual particles, the SD ranges between 0.07 and 0.10 ns, more than a factor of 2 smaller.

One possible cause for the heterogeneity in the sample is fluctuations in the synthesis conditions, as changing the temperature or the acidity during synthesis results in small, but noticeable changes in the lifetime (Supplementary Fig. 7a). These parameters critically affect the rate of crystallite formation and their grow, which in turn can lead to differences in particle size, morphology, and lattice regularity, properties that all affect the inherent defect level. If aggregates already form during synthesis, their constituent crystallites are all created under similar conditions and therefore there is little variation between them. Particles that formed in different regions of the synthesis vessel at different times, on the other hand, experience stronger variations in synthesis conditions, resulting in a broader distribution in functionalization and defect level. However, analysing the phasor plots of the individual samples more closely suggests the presence of two distinct species, where the majority of particles show a shorter lifetime, whereas a small fraction exhibits longer decays (Supplementary Fig. 7b). This distinction is preserved for the different synthesis conditions. We further investigated whether these two species are inherent to the MOF or are induced by the dye. For this, the emission of the unfunctionalized UiO-67 was measured. Even without external fluorophores, the MOF shows luminescence, but requires much higher excitation power

($\sim 100$–$1000 \times$) at a lower excitation wavelength (405 nm). Again, two populations can be distinguished, mostly via a difference in emission intensity (Supplementary Fig. 7c). This suggests that two species of MOF crystals are present that differ slightly in their properties. Although FLIM alone cannot determine the exact source of the observed differences, these results highlight the power of FLIM to detect small variations between different synthesis conditions and even within the sample, generally not possible with bulk measurements.

**Comparison of de novo and postsynthetic modification.** As the fluorescence lifetime can be used to measure the defect levels in MOFs, we can use this technique to compare de novo and linker exchange functionalization methods for incorporating modified linkers into the framework. Linker exchange is a form of post-synthetic modification, meaning that the functional groups are incorporated after the scaffold has been formed[21, 39]. Therefore, it should show a different propensity for defect formation. Three different samples were functionalized via linker exchange by heating pre-formed UiO-67 crystals to 65 °C for 1 h, 6 h or 24 h in the presence of excess dye-modified linkers (Fig. 4). The total number of incorporated FITC linker is nearly 1% of all linkers for all three exchange times, much higher than with de novo synthesis (Supplementary Table 1). An analysis of the intensity images revealed that individual MOF particles often appear as ring-like structures with a darker core at the centre (Fig. 4b–d). This heterogeneity can be attributed to the functionalization mechanism. During linker exchange, linkers labeled with dyes are added to the pre-formed MOFs and have to diffuse to the location in the framework where they exchange. The external surface of the crystals provides many CUS that are readily available for binding where the dye modified linkers do not even have to enter the pores[40]. Similar outside-in mechanisms have been previously observed for linker exchange and diffusion into MOFs[13, 41, 42]. Internal labelling is more difficult as the large size of the dye functionalized linkers relative to the pore and window diameters of UiO-67 make the diffusion process slow. Hence, a strong concentration gradient is created from the particles' exterior to their centre, especially for short linker exchange times. Together, these factors result in a distribution of the dyes close to the surface, reflected by the ring-like structures in the fluorescence images. For de novo functionalization, these rings are not observed, indicating that there the distribution is homogeneous throughout the crystal (Fig. 2 and Supplementary Fig. 5).

The fluorescence lifetime of the postsynthetically modified samples, on the other hand, is barely affected by the high dye concentration (Fig. 4 and Supplementary Table 1). The 1 h linker exchange MOF has an apparent lifetime of 2.11 ± 0.23 ns, very similar to the 2.28 ± 0.15 ns observed for the de novo sample with the lowest FITC concentration. The fact that the FITC concentration is more than 30 times higher but results in a similar lifetime strongly supports the hypothesis that energy transfer to defects, rather than between different fluorophores, is the main quenching mechanism for FITC. This difference in defect level is due to the fact that the MOF was first synthesized without any modification. As no dyes were present to interfere with the scaffold formation, the MOF exhibits fewer defects. Furthermore, once the framework is already fully formed, it is much harder to introduce defects that extend over nanometers. This makes the samples more resilient and the postsynthetic incorporation of the FITC modified linkers is less disruptive to the MOF backbone and less quenching is observed. Although short exchange times result in long fluorescence lifetimes, longer incubation times result in a shortening of the lifetime down to 1.40 ± 0.15 ns for an exchange time of 24 h (Fig. 4 and

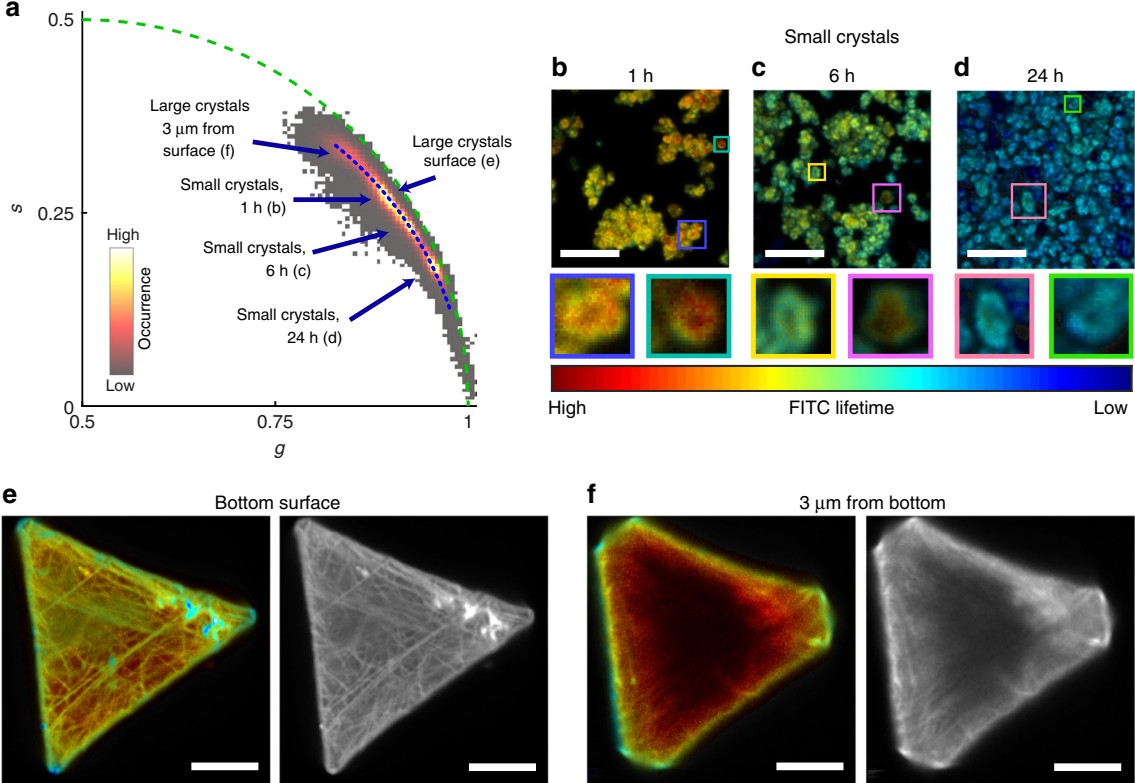

**Fig. 4** Phasor FLIM data of UiO-67 samples functionalized with FITC using linker exchange. **a** A phasor plot of the images shown in **b–f**. The blue arrows point at the average phasor positions of the different samples. **b–d** FLIM images of small crystal UiO-67 samples subjected to linker exchange with FITC-modified linkers at 65 °C for 1 h (**b**), 6 h (**b**), and 24 h (**c**). The inserts below are magnifications of the colored squares. **e**, **f** FLIM (in color) and fluorescence intensity (in gray scale) images of large crystal UiO-67 samples subjected to linker exchange with FITC-modified linkers for 24 h. **e** An image recorded at the bottom surface of a crystal. **f** An image recorded inside the crystal, 3 μm up from the bottom surface. The color coding of the images is based on the blue dotted line in panel **a** using the color table shown in **b**. The scale bar of all images is 10 μm

Supplementary Table 1). This suggests that more defects are present after linker exchange at 65 °C for 24 h, created by prolonged incubation at higher temperature in the presence of the dye. Other possible explanations for the shorter lifetime are an increase indirect dye–dye interactions or the redistribution of the fluorophores in the crystals. Both explanations are unlikely, as the amount of fluorophores does not change (Supplementary Table 1), whereas a redistribution means the dyes move toward the center of the crystals, where the fluorescence lifetime is expected to increase rather than decrease (Fig. 4b).

**Measuring internal heterogeneities**. With the de novo experiments, we observed that incorporation of the dye linkers into the scaffold results in the formation of defects, which, in turn, quench the fluorophores. The problem hereby is that the probe (i.e., dye) causes the defect itself rather than measuring the inherent level of defects. To perform experiments where the formation and the sensing of defect were decoupled, we used MOFs that were de novo functionalized with 0.1% or 2% FITC, and added RITC via postsynthetic treatment. Here, FITC was used to modulate the amount of defects and RITC was used to measure the defects in the crystal. RITC, this time without the linker, was added after the synthesis and diffused into the pores at 100 °C over 24 h. As both samples were treated identically, any differences in the measured RITC lifetime will be due to the different defect level introduced by the FITC.

The lifetime of the FITC signal is not affected by the incorporation of RITC (Supplementary Fig. 8a), showing that the postsynthetic treatment did not create additional defects. The

signal for RITC, on the other hand, shows a clear correlation between the FITC concentration and the RITC lifetime (Supplementary Fig. 8b). The lifetime decreases from 3.26 ± 0.14 ns for the 0.1% sample to 2.71 ± 0.15 ns for the 2% sample due to the higher defect concentration. The intensity distribution again shows the ring-like structures for the dye added postsynthetically, whereas the de novo functionalization results in a more even distribution, indicating a homogeneous incorporation throughout the crystals.

**Mapping chemical diversity within single crystals**. As the size of the individual crystallites is not much bigger that the resolution limit of the microscope, the borders between crystal surface and interior become blurred, decreasing the contrast in both fluorescence intensity and lifetime. To better distinguish these two areas, a UiO-67 sample was synthesized using an alternative protocol that yields significantly larger crystals that are ~ 30 μm in diameter (Fig. 4e, f and Supplementary Fig. 9)[43]. The MOF was then functionalized with FITC-modified linkers using linker exchange at 100 °C for 24 h. Fluorescence intensity images were taken at different heights, with 500 nm separation between the different planes, showing the three-dimensional intensity distribution of the crystals (Supplementary Fig. 9). The lifetime was measured at the bottom surface of the crystals (Fig. 4e). In addition, FLIM images were recorded 3 μm above the surface plane (Fig. 4f) to limit the influence of surface bound fluorescence (axial focus size ≈ 1 μm).

Just as with the small crystallites, this functionalization method resulted in high fluorescence intensity on the outer surface and a

gradual decrease of the signal toward the crystal interior (Supplementary Fig. 9). The edges, vertices, and cracks in the surface show even higher fluorescence intensity, indicating that these regions likely have a very high number of under-coordinated sites where the dye-modified linkers can easily bind. The fluorescence lifetime inside the crystals was found to be 2.73 ± 0.19 ns, significantly longer than for any of the other FITC-functionalized samples. The value for the surface plane (2.19 ± 0.17 ns), on the other hand, corresponds well to the lifetime of the small crystal MOF modified with linker exchange for 1 h (2.11 ± 0.23 ns), as both have a similar contribution from the surface. This difference in fluorescence lifetime can either be caused by an enhancement and stabilization of the fluorophores inside of the pores, or by quenching of the fluorescence at the external surface. As the fluorescence quantum yield of FITC is already close to unity, a further enhancement is unlikely. Surface induced quenching, on the other hand, is consistent with our observation of defects as quenchers, especially when considering that the edges and cracks show a faster decay compared with the more homogeneous surface areas (Fig. 4e, f).

We further investigated whether a difference between the surface and the interior of the crystal can be observed without adding fluorophores. For this, the pure UiO-67 crystals were imaged directly at the surface and 4 μm into the crystal (Supplementary Fig. 10). Again, the images show a clear difference between the external surface and bulk of the crystal, in both the fluorescence intensity as well as the lifetime, which is 5.08 ± 0.59 ns at the center and 3.49 ± 0.41 ns on the surface.

## Discussion

In this report, we have demonstrated how to use fluorescence imaging microscopy combined with lifetime analysis to resolve the chemical diversity of a MOF in three dimensions. Specifically, we investigated the functionalization chemistry of UiO-67 and its effect on the structural features of the scaffold. The sub-micrometer resolution of the microscope allowed us to identify heterogeneity between individual particles in a sample and even within single crystal, e.g., between the surface and the bulk. This difference between the outside and the inside was also observed for the auto-luminescence lifetime of UiO-67 in the absence of additional fluorophores, making the methods also applicable to the study of non-fluorescent functional groups.

Based on the results we can draw several conclusions about UiO-67 and its functionalization chemistry. The FRET experiments showed no direct dye–dye interactions, indicating that de novo functionalization of the framework results in a homogeneous random distribution of the functional groups. Even though no FRET signal was detected, higher dye concentrations still led to a decrease in fluorescence lifetime. This we attribute to the creation of nanoscale defects in the scaffold caused by the incorporation of the dyes. In addition, imaging revealed a high degree of diversity in each sample for both the concentration of the dyes and the level of defects. Individual crystallites in an aggregate, on the other hand, showed a much narrower distribution of these properties. Furthermore, we compared linker exchange incorporation of the fluorophores to de novo functionalization. These experiments showed that the postsynthetic modification results in fewer defects than an incorporation during the initial synthesis. This is even more pronounced for larger crystals that tend to be more resilient to defect formation then small crystallites.

These insights highlight the power and versatility of lifetime imaging in measuring and spatially resolving the chemical diversity in porous materials. The high sensitivity of the fluorescence lifetime to the local environment of the fluorophore makes it an ideal parameter to study a variety of properties, including defects, the presence of different functional groups, or the solvent present in the pores. In addition, one can exploit the auto-luminescence of the framework without the need to modify the material in any way. Thus, we believe that advanced fluorescence microscopy has great potential in the field of material science and that a wider application of the technique will lead to a wealth of new information.

## Methods

**Chemicals and supplies.** Anhydrous N,N-dimethylformamide (DMF), anhydrous methanol, and glacial acetic acid were obtained from EMD Millipore Chemicals. Ethyl acetate (99.9%, HPLC Plus), ethanol (≥ 99.5%, ACS Reagent), isopropyl alcohol (≥ 99.5%, BioReagent), tetrahydrofuran (THF, ≥ 99.9%, for high-performance liquid chromatography (HPLC)), nitric acid (70%, ACS Reagent), sulfuric acid (95.0–98.0%, ACS Reagent), hydrochloric acid (37%, ACS Reagent), sodium bicarbonate (BioReagent), sodium sulfate (≥ 99.0, ACS Reagent), palladium on carbon (Pd/C, 10 wt.% loading), potassium hydroxide (90%), zirconium chloride ((≥ 99.5%, trace metal basis), BPDC (97%), dimethyl biphenyl-4,4'-dicarboxylate (99%), fluorescein-5(6)-isothiocyanate (≥ 90%), and RITC (mixed isomers) were obtained from Sigma-Aldrich. Ultra-high-purity grade $H_2$ gas (Praxair, 99.999% purity) was used for the hydrogenation reaction. All starting materials, reagents, and solvents were used without further purification.

**Synthesis of dye-functionalized linkers.** The dye-modified linkers were synthesized in a multi-step procedure. The synthetic approach is illustrated in Supplementary Fig. 11.

Dimethyl 2-nitrobiphenyl-4,4'-dicarboxylate (**II**): A solution of 10 g (37 mmol) of dimethyl biphenyl-4,4'-dicarboxylate (**I**) in 100 ml of concentrated sulfuric acid was cooled to 0 °C and a mixture of 5 mL of 58% nitric acid and 7.5 mL of concentrated sulfuric acid was added dropwise over a period of 30 min under stirring, maintaining the temperature at 0–5 °C. The mixture was then stirred for 1 h at 0–5 °C, diluted with 100 mL of water, and extracted with ethyl acetate. The extract was washed with water and a solution of sodium bicarbonate (7.5%), dried over anhydrous sodium sulfate, and evaporated. The residue was recrystallized from isopropyl alcohol. Yield: 9.9 g (31 mmol, 85%). $^1$H NMR (400 MHz, CDCl$_3$), parts per million [p.p.m.]: 3.69 (s, 3 H), 3.94 (s, 3 H), 7.55 (d, 2 H), 7.73 (d, 1 H), 8.03 (d, 2 H), 8.27 (dd, 1 H), 9.97 (d, 1 H).

Dimethyl 2-aminobiphenyl-4,4'-dicarboxylate (**III**): A mixture of 9.9 g (31 mmol) of compound **II**, 100 mL of acetic acid, and 5 g of 10% Pd/C in a high-pressure reactor was hydrogenated at room temperature and a hydrogen pressure of 10–50 atm until hydrogen was no longer consumed. The mixture was filtered and acetic acid in the filtrate was removed under vacuum. The crude product was recrystallized from ethanol. Yield: 8.8 g (29 mmol, 94%). $^1$H NMR (400 MHz, CDCl$_3$), [p.p.m.]: 3.39 (s, 2 H), 3.83 (s, 3 H), 3.88 (s, 3 H), 7.17–8.09 (m, 7 H).

2-Aminobiphenyl-4,4'-dicarboxylic acid (NH$_2$-H$_2$BPDC, **IV**): A solution of 4.95 g (20 mmol), compound **III**, in THF (180 mL) and methanol (130 mL) was mixed with a solution of potassium hydroxide (10 g, 178 mmol) in water (200 mL). The reaction mixture was heated to reflux overnight. After all the volatiles were removed under vacuum, it was diluted with 200 mL of water, and acidified with 6 M hydrochloric acid until pH 2. The precipitates were collected, washed with water, and dried in air. The yield was 4.3 g (17 mmol, 85%). $^1$H NMR (400 MHz, dimethyl sulfoxide (DMSO)), [p.p.m.]: 5.15 (s, 2 H), 7.11 (d, 1 H), 7.21 (dd, 2 H), 7.41 (d, 1 H), 7.57 (m, 2 H), 8.01 (m, 2 H), 12.92 (s, 2 H).

2-Fluorescein-5(6)-isothiocyanate-biphenyl-4,4'-dicarboxylic acid (FITC-H$_2$BPDC, **VI**): A solution of 0.50 g (1.9 mmol) of compound **IV** in DMF (5 mL) was added to 0.75 g (1.9 mmol) of fluorescein-5(6)-isothiocyanate (FITC, **V**). The reaction solution was stirred for 24 h at room temperature. The mixture was then diluted with 100 mL of 1 M hydrochloric acid. The precipitates were collected by filtration, washed with water, and dried in air. The crude product was subject to purification by preparation HPLC (stationary phase: C18; mobile phase: methanol/water/0.1% trifluoroacetic acid (TFA). The eluent was freeze-dried and the orange solid was collected. The yield was 0.71 g (1.1 mmol, 58%). mass spectrometry (MS) (electrospray ionization (ESI-), m/z): [M-H]$^-$ calculated for $C_{35}H_{21}O_9N_2S^-$, 645.0973; found, 645.0956.

2-RhodamineB-isothiocyanate-biphenyl-4,4'-dicarboxylic acid (RITC-H$_2$BPDC, **VIII**): A solution of 0.25 g (0.95 mmol, **VII**), compound **IV**, in DMF (5 ml) was added to 0.51 g (0.95 mmol) of RITC (mixed isomers). The reaction solution was stirred for 24 h at room temperature. The mixture was then diluted with 100 mL of 1 M hydrochloric acid. The precipitates were collected by filtration, washed with water, and dried in air. The crude product was subject to purification by preparation HPLC (stationary phase: C18; mobile phase: methanol/water/0.1% TFA). The eluent was frozen-dried and the orange solid was collected. The yield was 0.30 g (0.36 mmol, 38%). MS (ESI-, m/z): [M-H]$^-$ calculated for $C_{45}H_{45}O_7N_4S^+$, 785.3003; found, 785.3005.

**Synthesis of small crystal UiO-67 and de novo functionalization.** The synthesis of pure and functionalized UiO-67 followed a modified synthesis protocol by Katz

et al.[23]. BPDC (H₂BPDC) and zirconium(IV) chloride (18.2 mg, 78.1 μmol) were mixed in a 4 mL scintillation vial. In case of de novo functionalized UiO-67, some of the linker was replaced with 2-amino-BPDC modified with either FITC (FITC-H₂BPDC) or rhodamineB isothiocyanate (RITC-H₂BPDC) (see Supplementary Table 3 and Supplementary Fig. 11). Dimethylformamide (DMF, 2 mL) and hydrochloric acid (0.1 mL, 37%) were added to this mixture. Next, the mixture was sonicated for 20 min. The vial was then heated at 90 °C for 24 h, yielding functionalized, small crystal UiO-67. The sample was then washed by immersing it in 4 mL of anhydrous DMF for 3 days. During this time, the DMF was replaced five times per day. The washing procedure was subsequently repeated with anhydrous methanol. The methanol exchanged sample was then evacuated at room temperature under vacuum for 24 h.

**Synthesis of large crystal UiO-67.** Large UiO-67 single crystals are synthesized following the procedure of Ko et al.[43]. A mixture of H₂BPDC (85 mg, 0.35 mmol), zirconium(IV) chloride (82 mg, 0.35 mmol), and benzoic acid (1.28 g, 10.5 mmol) was dissolved in DMF (20 mL) in a 20 mL vial. The vial was capped and heated in an isothermal oven at 120 °C for 2 days to yield octahedral-shaped crystals of ~ 30 μm diameter. The reaction mixture was allowed to cool down to room temperature and then washed with DMF (three times per day for 3 days) and acetone (three times per day for 3 days). The solvent exchanged samples were then evacuated at 120 °C to 30 mTorr.

**Linker exchange functionalization.** Unfunctionalized UiO-67 (12.5 mg) was placed in a 20 mL scintillation vial and a solution of FITC-H₂BPDC (2.0 mg, 3.1 μmol) in DMF (20 mL) was added. The mixture was heated to 65 °C or 100 °C, and at certain time points (1 h, 6 h, 24 h) a fraction of the suspension was removed. The extracted samples were centrifuged and the solid was subsequently washed with anhydrous DMF (10 times, 4 mL each) and anhydrous methanol (10 times, 2 mL each). The washed samples were then dried under vacuum for 24 h.

**Dye diffusion incorporation.** The de novo functionalized samples with 0.1 and 2% FITC or RITC were further functionalized by letting free dye (RITC for FITC-linker functionalized samples and FITC for RITC-linker functionalized samples) diffuse into the pores. For this, a suspension of the MOFs (0.625 mg mL⁻¹) in DMF with the free dye (0.58 mM) were heated to 100 °C for 24 h. Subsequently, the samples were washed three times each with DMF (1 mL) and methanol (1 mL) and dried at 70 °C in an oven.

**Dye concentration measurements.** The fraction of dyes in the MOF samples was determined via fluorometry. Hereby, aliquots of the respective samples (0.3–0.6 mg) were digested in a solution of cesium fluoride (137.2 mg, 0.90 mmol) in a mixture of water (1.43 mL) and DMSO (2.57 mL). The samples were sonicated for 10 min and subsequently incubated at room temperature for 2 h, to ensure a full digestion of the framework. The stock solutions (3 mL) were then measured in a polymethylmethacrylat cuvette on a fluorescence system consisting of an 814 Photomultiplier Detector, a LPS220B Lamp Power Supply, a Pti-MD3020 Motor Drive (all Photon Technology International) and a TC125 Temperature control (Quantum Northwest). For calibrating the conversion factors from the fluorescence signal to the concentration, different concentrations of the raw modified linker molecules were measured under identical conditions and fit with a linear function. Based on the measured dye concentration and the amount of dissolved MOF, the fraction of linkers modified with a dye were calculated in p.p.m.

**Nearest-neighbor calculation.** To estimate how far apart the fluorophores should be from each other, on average, assuming a perfectly random distribution in the MOFs, the nearest-neighbor distance was calculated according to:

$$P(r) = \frac{3}{a}\left(\frac{r}{a}\right)^2 e^{-\left(\frac{r}{a}\right)^3}, \text{ with } a = \left(\frac{3}{4\pi n}\right) \tag{1}$$

Here, $P(r)$ is the probability of finding the nearest fluorophore at distance $r$. The average dye density $n$ corresponds to the number of dye molecules, $N$, per volume, $V$, and is calculated as:

$$n = \frac{N}{V} = \frac{24 \cdot c}{(26.783\text{Å})^3} \tag{2}$$

where $c$ is the fraction of linkers modified with a dye. The number 24 represents the number of linkers per unit cell with a size of 26.783 Å.

Based on this, we obtain a mean nearest neighbour distance $\langle r \rangle$ of:

$$\langle r \rangle = a \cdot \left(\frac{4}{3}\right) \tag{3}$$

with $\Gamma$ representing the gamma function.

**FLIM microscopy.** All fluorescence lifetime and intensity images were recorded on a home-built laser scanning confocal microscope equipped with pulsed interleaved

excitation and time-correlated single photon counting detection, as described previously[44]. For the measurements, 20–30 μL of a suspension of the MOFs in water (1–10 mg mL⁻¹) were placed in an 8-Well LabTek I slide (VWR). Once the particles sedimented, the surface was imaged using a ×60, 1.27 numerical aperture water-immersion objective (Plan Apo IR ×60 WI, Nikon). The resolution was set to 300 by 300 pixels, resulting in a pixel size of 100 nm (30 μm total image size) or 333 nm (100 μm total image size). To ensure a good signal to noise ratio while, at the same time, minimize the influence of photon pile-up and other high signal artifacts, the count rate was kept between 50 and 500 kHz. To achieve this, the laser power (475 nm and 565 nm for FITC and RITC, respectively) was adjusted in a range of 1–20 nW for dye-functionalized samples, as measured before the objective. For autoluminescence measurements in the absence of fluorophores, a 405 nm laser was used at a power of 10 μW. Under these conditions with a total measurement time of 250–500 s, this resulted in 200–5000 photons per pixel. All analysis was performed using the software framework PAM[45].

**Phasor approach to fluorescence lifetime analysis.** As there are many possible contributions to fluorescence quenching in MOFs, a fit-based quantitative analysis is difficult to perform and can even be biased when an inappropriate fit model is used. The phasor approach to FLIM[26, 27], on the other hand, uses the Fourier space to visualize the measured fluorescence lifetime in a graphical way. The phasor is calculated via simple equations and is therefore not based by any fit models. This makes it very useful for a qualitative analysis of complex data with many unknown processes and contributions, as is the case for the presented data.

For each pixel, the Fourier coordinates, $g$ and $s$, are calculated using equations 4 and 5:

$$g_{i,j}(\omega) = \int_0^{2\pi} I_{i,j}(t) \cdot \cos(\omega t - \varphi_{\text{Inst}})\mathrm{d}t \Big/ \left( M_{\text{Inst}} \cdot \int_0^{2\pi} I_{i,j}(t)\mathrm{d}t \right) \tag{4}$$

$$s_{i,j}(\omega) = \int_0^{2\pi} I_{i,j}(t) \cdot \sin(\omega t - \varphi_{\text{Inst}})\mathrm{d}t \Big/ \left( M_{\text{Inst}} \cdot \int_0^{2\pi} I_{i,j}(t)\mathrm{d}t \right) \tag{5}$$

Here, the indices $i$ and $j$ define the pixel coordinates in the image and $I(t)$ gives the photon counts of the time bin $t$. The frequency $\omega$ corresponds to $2\pi/T$, with $T$ being the full range of the photon arrival time histogram (here 40 ns). The correction terms, $\varphi_{\text{Inst}}$ and $M_{\text{Inst}}$, account for the instrument response function (IRF) and can be calculated by measuring a reference sample with known lifetime (here Atto488 4.1 ns, Atto-Tec).

There are three main rules of the phasor space that simplify interpretation of lifetime data:

1. A convolution of a decay with a different signal (e.g., the IRF) results in a change of the coordinate system. This makes it possible to use $\varphi_{\text{Inst}}$ and $M_{\text{Inst}}$ to correct for the IRF without complicated convolutions.
2. All purely mono-exponential decays lie on the semi-circle of radius 0.5 cantered at (0.5,0). The lifetimes decrease clockwise from infinity at the origin to zero at the point (1,0).
3. A mixture of different lifetimes (i.e., bi- or multi-exponential decays) results in a phasor that is the weighted vector addition of the phasors of the base components. This results in the fact that any mixture of two phasors lie on a straight line connecting the original phasors. For more components, the possible space of the mixture is a polygon with the phasors of the original components at the vertices. As a consequence, all multi-exponential decays must fall inside the semi-circle.

From rules 2 and 3, it follows that curved trajectories are caused by a gradual change in the components' lifetimes, rather than a change in the relative contribution of species with constant lifetimes.

**Quantitative lifetime analysis.** For each phasor position, two lifetime values can be calculated based on the phase ($\tau_\varphi$) and the modulation ($\tau_M$):

$$\tau_\varphi(\omega) = \frac{1}{\omega} \cdot \frac{s}{g} \tag{6}$$

$$\tau_M(\omega) = \frac{1}{\omega}\sqrt{\frac{1}{g^2 + s^2} - 1} \tag{7}$$

For purely mono-exponential decays, these two lifetimes are identical and correspond to the real lifetime. In the case of multi-exponential behavior, the phase and modulation lifetimes are different and do not correspond directly and unambiguously to the pure components.

To get a single apparent lifetime for each sample, first the mean $\tau_\varphi$ and $\tau_M$ are calculated from all pixels above a threshold of 300 photons. The arithmetic average of the mean phase and modulation lifetimes is then used to calculate an apparent lifetime. The uncertainty corresponds to the SD of the pixel distribution.

**Powder X-ray diffraction.** PXRD measurements were conducted on a Bruker D8-Venture diffractometer with a Mo-target (0.71073 Å) and Cu-target (1.54184 Å) microfocus X-ray generators. The $\theta$–$\theta$ geometry device was equipped with a PHOTON-100 CMOS detector, measuring the samples between 2° and 50° $2\theta$, with a step size of 0.02° of $2\theta$.

**Nitrogen adsorption/desorption isotherms.** Gas adsorption analysis was performed on a Quantachrome Quadrasorb-SI automatic volumetric gas adsorption analyser. A liquid nitrogen bath (77 K) and ultrahigh purity grade $N_2$ (99.999%, Praxair) were used for the measurements. Samples were prepared and measured after being evacuated at 100 °C for 12 h. In order to calculate pore size and volume, calculations were performed using a slit-pore based $N_2$ on carbon QSDFT equilibrium model. To calculate the Brunauer–Emmett–Teller surface area a partial pressure range between 0.05 and 0.15 $p/p_0$ was used.

**Scanning electron microscopy.** A Zeiss NVision40 microscope was used to record SEM images. Secondary electron images were acquired using the InLens detector at a low acceleration voltage of 5 kV. To avoid charging effects, a thin carbon film coating was applied on the samples before the measurements. The carbon deposition was performed using a BAL-TEC coating system.

**Data availability.** The authors declare that all data supporting the findings of this study are available within the article and its Supplementary Information or from the corresponding author upon reasonable request.

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

# Acknowledgements

We thank Nader Danaf (LMU) for performing a few additional experiments. We are grateful for financial support from the Deutsche Forschungsgemeinschaft (DFG) through

DFG-project WU 622/4-1 (W.S., P.H., and S.W.) and the SFB1032 (Project B3, D.C.L.). We also thankfully acknowledge the support of the Excellence Cluster Nanosystems Initiative Munich (NIM), the Center for NanoScience Munich (CeNS), and the LMUinnovativ BioImaging Network.

## Author contributions

W.S. performed and analysed the microscopy experiments. J.J. and Z.J. synthesized and functionalized the samples. J.J. and P.H. performed and analysed the bulk experiments. O.M.Y. and S.W. designed the project and led it together with D.C.L. W.S., Z.J., O.M.Y., and S.W. prepared the first version of the manuscript and all authors contributed to the final version.

## Additional information

**Competing interests:** The authors declare no competing interests.

