## [Peer Review File · Nature Communications]

Reviewers' comments:

Reviewer #1 (Remarks to the Author):

In this paper, the authors introduce fluorescent probes in to single crystal of UiO-67, through (i) de novo synthesis and (ii) post-synthetic modification. Fluorescence lifetime imaging is then used in identifying that dyes reside mainly on the surfaces of the crystal in (ii), and are distributed throughout (not phase separated) the crystal in (i). The author also shows that the introduction of dye molecules introduces defects.

The authors do additional experiments to decouple defect introduction and dye incorporation. They also show that the introduction of dye molecules to use FLIM are not necessary at all, and that a lower wavelength and higher excitation power can yield results. The work is however carried out competently well.

The attempts made by the authors to address my first concerns are solid, and considerable times and effort has been spent doing extra experiments. Major concern remains however, which are not questioning the science presented, but rather the motivations and generalized conclusions for this technique.

1. In my previous report, I stated that “the authors should be trying to use a technique to visualize the spatial distribution of inherent defects or functional site ordering within a MOF, not imaging the defects/functional group distribution caused by the technique-enabling molecules introduced”. In this revised version, the decoupled experiments still show that the first dye is causing defects in the crystal. The second then senses these. My original comment was why the others are introducing defects/dyes into the crystal in the first place, rather than just looking at using the technique with a as synthesized UiO-67 composition. The result also shows that samples from the same batch of synthesis contain different morphologies or crystal environments.
2. The authors have then answered this point, late in the paper, with an excellent study of pure UiO-67 crystals using FLIM, to show that there is a difference between crystal and surface. This is however a very minor part of the paper – but if dyes are not necessary then is the main message of the paper that the introduction of dyes causes defects which can be measured, but that the use of FLIM doesn't require this? In which case, I cannot see the benefit of the study.
3. The previous work by Wuttke et al (Ref 17) uses phasor FLIM with MIL-101, to show that the same batch can contain different crystal morphologies. Part of the additional experiments look to be a replication of that with UiO-67. The last sentence before the results and discussion shows that similar, if not identical conclusions (i.e. one batch yields at least two different MOF crystal(lite) populations) were reached. I now cannot see much of a difference between the two papers, and the authors could have been clearer in delineating them.

The paper would have been, in my opinion, of a significantly higher impact had the application of FLIM without dye incorporation to MOFs generated information not observable by another method, or at least agreed with methods significantly harder to implement. As it is, it is very confusing that most of the paper concentrates on introducing dyes (in order to use FLIM), which causes modification of synthesized UiO-67, which is found to be unnecessary.

Reviewer #2 (Remarks to the Author):

The authors have clearly done a lot of work on this study since the original review, and they have done it well. I still believe this to be an excellent piece of work, and recommend publication in Nature Communications. The methodology described to study metal-organic frameworks in detail will be of interest to a wide audience, including both the extensive MOF and photochemical communities.

All the original concerns have been addressed in detail, and I have only the following minor comments and queries based on the current version of the manuscript and the authors' responses:

1. In response to Reviewer #1, points C&D, the authors describe three new experiments. In experiment 2 they describe the MOF syntheses carried out with small condition variations. This set of experiments is excellent, but I may have missed where the effect of varied conditions on particle size/morphology is discussed? Changes in the particle size and shape properties may be a contributing factor to the observed changes in fluorescence lifetime between samples.

2. Response to Reviewer #2 point 8: I completely understand the reluctance to introduce a more quantitative analysis of the fluorescence lifetimes than the phasor approach described here, but am still a little disappointed that the intrinsic information about donor-quencher distances is not further explored – perhaps this is something that could be investigated more in future work. With good computational support, models of defect/donor distributions and relative orientations could be extracted from experiments such as these.

3. Response to Reviewer #3, second paragraph: I am not convinced that existing studies focusing on “accessibility of... ..active sites” is as conceptually far from using the authors' approach to interrogating “defects” as the authors suggest here. This is just my observation, and does not affect the manuscript text at all.

4. Manuscript, pages 11 and 12: The additional experiments using small variations in synthetic conditions are very good and the outcomes are fascinating. The observation of two species of MOF crystals of UiO-67 is particularly interesting, and certainly demonstrates one of the strengths of this analytical approach.

5. Page 13, line 4: The addition of the previous new section has made the introductory sentence to this paragraph a little unclear, perhaps it could be rephrased to clarify the sample under discussion. (The fluorescence lifetime of 'what', precisely?).

6. Page 16, line 9: Should this read 3 μm into the crystal rather than 4 μm ? The text accompanying Figure S10 says 3 μm .

I look forward to seeing further work along these lines in the future as there is much to discover about many MOFs using methods like these.

Response to Reviewers' comments:

Reviewer #1:

In this paper, the authors introduce fluorescent probes in to single crystal of UiO-67, through (i) de novo synthesis and (ii) post-synthetic modification. Fluorescence lifetime imaging is then used in identifying that dyes reside mainly on the surfaces of the crystal in (ii), and are distributed throughout (not phase separated) the crystal in (i). The author also shows that the introduction of dye molecules introduces defects.

The authors do additional experiments to decouple defect introduction and dye incorporation. They also show that the introduction of dye molecules to use FLIM are not necessary at all, and that a lower wavelength and higher excitation power can yield results. The work is however carried out competently well.

The attempts made by the authors to address my first concerns are solid, and considerable times and effort has been spent doing extra experiments. Major concern remains however, which are not questioning the science presented, but rather the motivations and generalized conclusions for this technique.

1. In my previous report, I stated that “the authors should be trying to use a technique to visualize the spatial distribution of inherent defects or functional site ordering within a MOF, not imaging the defects/functional group distribution caused by the technique-enabling molecules introduced”. In this revised version, the decoupled experiments still show that the first dye is causing defects in the crystal. The second then senses these. My original comment was why the others are introducing defects/dyes into the crystal in the first place, rather than just looking at using the technique with a as synthesized UiO-67 composition. The result also shows that samples from the same batch of synthesis contain different morphologies or crystal environments.

In general, the method we present here uses the incorporation of fluorescent markers with a phasor-FLIM approach in order to investigate the chemical diversity of MOFs. In the case of UiO-67, however, our in-depth analysis revealed that incorporation of fluorophores leads to defects in the crystal. This was not the result that we initially expected, but it gave us the opportunity to use the affect to both generate defects and investigate them in a systematic way.

The main focus of our paper is to use advanced fluorescence microscopy to reveal insights into the **chemical diversity in a MOF**. Our report is an in-depth investigation of the incorporation of functionalities (e.g. fluorescence dyes) *via* different functionalization approaches into a MOF **visualizing the dye distribution** as well as the **consequence for the structural features** of the MOF. With our Förster resonance energy transfer (FRET) experiments, we could determine that the arrangement of the incorporated functionalities was randomly distributed in the framework without any clustering or phase separation. We also observed a correlation between fluorescence lifetime and defects at the molecular level. With this correlation established, fluorescence imaging combined with lifetime analysis were employed to systematically map the compositional variation and defect distribution in multivariate UiO-67 with three-dimensional resolution.

In the revised version of the manuscript, we have highlighted the focus more clearly.

2. The authors have then answered this point, late in the paper, with an excellent study of pure UiO-67 crystals using FLIM, to show that there is a difference between crystal and surface. This is however a very minor part of the paper – but if dyes are not necessary then is the main message of the paper that the introduction of dyes causes defects which can be measured, but that the use of FLIM doesn't require this? In which case, I cannot see the benefit of the study.

We used the inherent luminescence of UiO-67 to demonstrate that introduction of fluorophores into the MOFs are not necessary. However, it would be dangerous to interpret too much into this data as

the source of the auto-luminescence signal is unknown. Hence, it is still necessary to complement the results measured using the inherent luminescence with experiments using fluorophores where the local influences on the fluorescence are better understood.

3. The previous work by Wuttke et al (Ref 17) uses phasor FLIM with MIL-101, to show that the same batch can contain different crystal morphologies. Part of the additional experiments look to be a replication of that with UiO-67. The last sentence before the results and discussion shows that similar, if not identical conclusions (i.e. one batch yields at least two different MOF crystal(lite) populations) were reached. I now cannot see much of a difference between the two papers, and the authors could have been clearer in delineating them.

The mention manuscript is a method paper showing for the first time the combination of fluorescence lifetime image microscopy (FLIM) and scanning electron microscopy (SEM). As the reviewer correctly remark, we investigated the correlation between crystal morphologies and observed quenching. However, the present paper focuses on using fluorescence imaging combined with lifetime analysis to reveal insight into the **chemical diversity in a MOF**. We show various aspect of the heterogeneity in UiO-67 (distribution of functional groups, measuring the chemical diversity between particles; comparison of *de novo* and post-synthetic modification, measuring internal heterogeneities, and mapping chemical diversity within single crystals).

The paper would have been, in my opinion, of a significantly higher impact had the application of FLIM without dye incorporation to MOFs generated information not observable by another method, or at least agreed with methods significantly harder to implement. As it is, it is very confusing that most of the paper concentrates on introducing dyes (in order to use FLIM), which causes modification of synthesized UiO-67, which is found to be unnecessary.

We agree that the paper would have been conceptually easier to follow had incorporation of the fluorophore not influenced the structure of the MOFs. However, as pointed out above, this effect turned out to be an opportunity for us to systematically adjust and monitor the amount of modifications added to the MOFs and investigate what effect the modifications had on the local structure of the MOF. The phasor-FLIM method has proven to be very powerful and can provide insights that are not easily obtainable with other methods. Besides highlighting the potential of FLIM in studying **chemical diversity in a MOF**, the manuscript also uncovers more specific aspects of UiO-67 chemistry:

- 1) *de novo* functionalization of the UiO-67 framework results in a homogeneous random distribution of the functional groups (e.g. dyes);
- 2) incorporation of the dyes creates nano-scale defects in the scaffold that can be detected *via* a decrease in fluorescence lifetime;
- 3) individual crystallites in an aggregate show less diversity than the full sample;
- 4) linker exchange incorporation of the fluorophores introduces fewer nano-scale defects compared to *de novo* functionalization;
- 5) larger crystals tend to contain fewer defects than small crystallites.

Thus, we believe the current results and the paper are of high impact to the MOF and porous material community.

At the end we would like to thank the reviewer for all his/her valuable comments that have clearly improved our manuscript.

Reviewer #2 (Remarks to the Author):

The authors have clearly done a lot of work on this study since the original review, and they have done it well. I still believe this to be an excellent piece of work, and recommend publication in Nature Communications. The methodology described to study metal-organic frameworks in detail will be of interest to a wide audience, including both the extensive MOF and photochemical communities.

All the original concerns have been addressed in detail, and I have only the following minor comments and queries based on the current version of the manuscript and the authors' responses:

1. In response to Reviewer #1, points C&D, the authors describe three new experiments. In experiment 2 they describe the MOF syntheses carried out with small condition variations. This set of experiments is excellent, but I may have missed where the effect of varied conditions on particle size/morphology is discussed? Changes in the particle size and shape properties may be a contributing factor to the observed changes in fluorescence lifetime between samples.

Thank you for pointing this oversight out to us. We have added a short discussion about possible changes in size/morphology to the manuscript.

2. Response to Reviewer #2 point 8: I completely understand the reluctance to introduce a more quantitative analysis of the fluorescence lifetimes than the phasor approach described here, but am still a little disappointed that the intrinsic information about donor-quencher distances is not further explored – perhaps this is something that could be investigated more in future work. With good computational support, models of defect/donor distributions and relative orientations could be extracted from experiments such as these.

We agree that a more quantitative analysis of the lifetime components can bring great insights into the nature and distribution of the defects. However, as the reviewer points out, such quantitative analysis would require “good computational support, models of defect/donor distributions and relative orientations”. A vast amount of effort would need to be invested to establish a quantitative relationship between the amount of quenching and distance to the quencher, which would probably require the involvement of groups that are specialized in computational analysis and modelling. Hence, these investigations would go well beyond the scope of the current manuscript and would potentially only contribute marginally to the main focus of this work.

3. Response to Reviewer #3, second paragraph: I am not convinced that existing studies focusing on “accessibility of... active sites” is as conceptually far from using the authors' approach to interrogating “defects” as the authors suggest here. This is just my observation, and does not affect the manuscript text at all.

We never intended to suggest that our approach is a completely new concept compared to previous studies, rather that it is a new perspective on the potential of the method and the insight that you can get with it.

4. Manuscript, pages 11 and 12: The additional experiments using small variations in synthetic conditions are very good and the outcomes are fascinating. The observation of two species of MOF crystals of UiO-67 is particularly interesting, and certainly demonstrates one of the strengths of this analytical approach.

This observation of the two species was a direct result of the useful suggestions for new experiments by the reviewers. While the two species were already present in the data of the original experiments,

the additional experiments made it very clear that this is a systematic property and not just some statistical occurrence. We thank the reviewer for her/his constructive comments.

5. Page 13, line 4: The addition of the previous new section has made the introductory sentence to this paragraph a little unclear, perhaps it could be rephrased to clarify the sample under discussion. (The fluorescence lifetime of 'what', precisely?).

We thank the reviewer for pointing out the confusion. We have adjusted the text to make this transition clearer.

6. Page 16, line 9: Should this read 3 μm into the crystal rather than 4 μm ? The text accompanying Figure S10 says 3 μm .

This was an error in the figure. It should have been 4 μm as is written in the text. Thank you for noticing this error. We have changed the figure accordingly.

I look forward to seeing further work along these lines in the future as there is much to discover about many MOFs using methods like these.

We thank the reviewer for these encouraging words and we greatly appreciate his positive feedback and helpful comments.

REVIEWERS' COMMENTS:

Reviewer #1 (Remarks to the Author):

The authors have responded to my comments without substantially changing the paper. The focus makes it slightly more impactful, and the science behind the technique development is interesting. The changes made in response to referee 2, who was much more positive, have also improved the manuscript. I have no further comments on this manuscript.